# Characterization of Ginsenosides from the Root of *Panax ginseng* by Integrating Untargeted Metabolites Using UPLC-Triple TOF-MS

**DOI:** 10.3390/molecules28052068

**Published:** 2023-02-22

**Authors:** Yizheng Sun, Xiaojie Fu, Ying Qu, Lihua Chen, Xiaoyan Liu, Zichao He, Jing Xu, Jiao Yang, Wen Ma, Jun Li, Qingmei Guo, Youbo Zhang

**Affiliations:** 1State Key Laboratory of Natural and Biomimetic Drugs, Department of Natural Medicines, School of Pharmaceutical Sciences, Peking University, Beijing 100191, China; 2School of Pharmacy, Shandong University of Traditional Chinese Medicine, Jinan 250355, China; 3Key Laboratory of Chemical Biology of Ministry of Education, Department of Natural Product Chemistry, School of Pharmaceutical Sciences, Shandong University, Jinan 250012, China; 4Department of Toxicology, School of Public Health, Peking University, Beijing 100191, China

**Keywords:** *Panax ginseng* meyer, ginsenosides, UPLC-Triple-TOF-MS/MS, growing environment

## Abstract

To compare the chemical distinctions of *Panax ginseng* Meyer in different growth environments and explore the effects of growth-environment factors on *P. ginseng* growth, an ultra-performance liquid chromatography–tandem triple quadrupole time-of-flight mass spectrometry (UPLC-Triple-TOF-MS/MS) was used to characterize the ginsenosides obtained by ultrasonic extraction from *P. ginseng* grown in different growing environments. Sixty-three ginsenosides were used as reference standards for accurate qualitative analysis. Cluster analysis was used to analyze the differences in main components and clarified the influence of growth environment factors on *P. ginseng* compounds. A total of 312 ginsenosides were identified in four types of *P. ginseng*, among which 75 were potential new ginsenosides. The number of ginsenosides in L15 was the highest, and the number of ginsenosides in the other three groups was similar, but it was a great difference in specie of ginsenosides. The study confirmed that different growing environments had a great influence on the constituents of *P. ginseng*, and provided a new breakthrough for the further study of the potential compounds in *P. ginseng*.

## 1. Introduction

Panax ginseng (the roots and rhizomes of *P. ginseng* Meyer, Family Araliaceae), first recorded in Shennong’s Herbal Medicine, contain a variety of chemical components as saponins, volatile oil, amino acids, alkaloids, vitamins, flavonoids, and organic acids [1,2,3]. Based on the chemical components, ginsengs can be used in food, dietary supplements, medicine, and other industries, and more and more *P. ginseng* appeared in the markets as a raw material of nutrition [4]. Clinically, *P. ginseng* plays an important role in the treatment of various diseases due to its abundant pharmacological activities as anti-oxidation [5], anti-diabetes, blood pressure regulation, anti-tumor [6], antimicrobial [7], relieving fatigue [8], and anti-inflammation [9,10].

As the main chemical components, ginsenosides play a key role in the pharmacological effects of *P. ginseng*. Up until now, more than 100 saponins have been isolated and identified from *P. ginseng* [11]. Ginsenosides can be roughly divided into three types: I: protopanaxadiol saponins (PPD), including Ra_1_, Ra_2_, Rb_1_, Rb_2_, and others; II: protopanaxatriol saponins (PPT), including Re, Rf, Rg_1_ and others; III: oleanolic acid (OA), such as Ro. In addition, there are ginsenosides with PPT-type side chain changes (PPTC) such as F4, ginsenosides with PPD-type side chain changes (PPDC) such as Rs4, Pseudoginsenoside (PG) such PG-F11. As the index components of *P. ginseng*, ginsenosides have been used as the quality standards nowadays for ginseng quality evaluation in multinational pharmacopeias or some local pharmaceutical guides.

Due to the maximum ultraviolet absorption at 190–200 nm, it is difficult to simultaneously detect hundreds of ginsenosides by the previous detectors, especially for the trace components. There are many analysis methods for ginsenosides, such as solid-phase methylation, electron-induced dissociation mass spectrometry, and UPLC-MS. However, there are many problems including poor analytical methods, low detection range of compounds, and low sensitivity. Ginsenosides of low content cannot be completely characterized, which means the chemical composition of ginseng cannot be studied in depth [12,13,14,15]. UPLC-Triple-TOF-MS/MS has the advantages of high-resolution mass spectrometry and triple quadrupole mass spectrometry [16,17], in which most possible precursor and fragment ions can be recorded synchronously. Other advantages of this instrument are high scanning speed, high resolution, high sensitivity, and high accuracy. This instrument was used in many fields of metabolomics analysis and natural product structure analysis [18,19].

Nowadays, with the improvement of living standards, the demand for ginseng has grown rapidly in recent years, which led to a large increase in the area of ginseng cultivation. However, not all areas can provide a quality living environment for the drug. To analyze the saponins of *P. ginseng* growing in different environments and preliminarily explore the influence of growth environment factors on the quality of *P. ginseng*, a qualitative determination method of saponins based on UPLC-Triple-TOF-MS/MS was established in this study. As a result, more than 300 ginsenosides including 75 potential new ginsenosides were identified in the *P. ginseng* growing in four different environments, and environmental factors were revealed to be an important role in the composition of ginsenosides in *P. ginseng*. The growth age is positively correlated with the number of ginsenosides in the root system of ginseng, which directly leads to the specificity and diversity of ginsenosides. The humid, low temperature environment is conducive to the diversity of total ginsenosides.

## 2. Results and Discussion

### 2.1. Analysis of the Structure of Ginsenosides

Accurate molecular weight and corresponding chemical formula can be obtained from primary mass spectrometry. MS/MS analysis was performed in a data-dependent acquisition (DDA) mode to produce a pattern of fragment masses unique to the single compound’s fragmentation. Thus, even if two compounds have the same retention time, they can be distinguished by their MS/MS spectra. On the basis of previous studies, compound identification was performed. A total of 312 ginsenosides, including the 63 standard compounds, were finally detected. Among the 312 ginsenosides, 237 are known compounds (Appendix A) [20,21,22,23,24,25,26,27,28,29,30,31,32,33,34,35,36,37,38,39,40,41,42,43,44] and 75 are potential new ginsenosides (Appendix A). A total of 63 ginsenosides were identified by comparison the references standards, 237 ginsenosides were assigned by comparison of the precise molecular weights, molecular formulae, the types of aglycones, glycosyl groups, chemical structures, and fragmental pathway with those in the literature. The type of aglycone and glycone are shown in Figure 1. There were 75 new potential ginsenosides, identified through the following steps: (a) the molecular formula of the deduced compounds was imported into the website of SCI-Finder to further confirm their structures; (b) compared with the searched compounds, if there is a consistent chemical group (regardless of the connection order of each group), the imported compound is considered to be known, otherwise, the ginsenoside is considered to be a new potential ginsenoside. According to MS and MS/MS data of reference compounds, some fragment ions were used as criteria to identify aglycons of the compounds. The unique fragment ions at *m*/*z* 487 [aglycone−H]^−^ (C_31_H_51_O_4_), 359 (C_24_H_39_O_2_), 31 (OCH_3_), 127 (C_7_H_11_O_2_, detached side-chain fragment) were evidence of PPDV3 ginsenosides. In the same way, the ions at *m*/*z* 441 [aglycone−H]^−^ (C_30_H_49_O_2_), 305 (C_20_H_34_O_2_) were evidence of aglycones PPDV1/V2, which were regarded as a product of H2O-removed PPD at C_20_–C_21_ or C_20_–C_22_. PPD and PPDV4 ginsenosides shared the same [aglycone−H]^−^ ion at *m*/*z* 459 (C_30_H_51_O_3_) and fragmental ion at *m*/*z* 305 (C_20_H_33_O_2_). The difference between them was the ion at *m*/*z* 373 (C_25_H_41_O_2_) and a neutral loss of 88 (C_5_H_12_O), which presented only in PPDV4 ginsenosides. However, the ion at *m*/*z* 373 was too weak in some PPDV4 ginsenosides to distinguish them from the PPD-type compounds. In addition, aglycones of PPTV1/V2, considered as H2O-removed PPT, would produce ions at *m*/*z* 457 [aglycone−H]^−^ (C_30_H_49_O_3_) and 321 (C_20_H_34_O_3_). Aglycones of PPTV3 and PPTV4, a pair of configuration isomer and additive products of PPT and H_2_O, showed unique ions at *m*/*z* 493 [aglycone−H]^−^ (C_30_H_53_O_4_) and 321 (C_20_H_34_O_3_). Aglycone of PPTV5, an isomer of hydroxylated PPTV1−type aglycone, had fragment ions at *m/z* 473 [aglycone−H]^−^ (C_30_H_49_O_4_) and 321 (C_20_H_34_O_3_). Thus, *m*/*z* 475 (C_30_H_51_O_4_) and 391 (C_26_H_39_O_4_) were the feature fragment ions of PPT−type aglycone. Aglycone of PG showed unique [aglycone−H]^−^ at *m*/*z* 491 (C_30_H_51_O_5_) and another ion at *m*/*z* 373 (C_25_H_41_O_2_), while aglycone of OA produced fragment ions at *m*/*z* 247 (C_15_H_37_O_2_) and 207 (C_14_H_23_O_1_) due to RDA cleavage. The structural classification of new compounds is shown in Figure 2. There are 46 of PPT-type, 23 of PPD-type, 5 of PG-type, and 1 of OA-type ginsenosides, and most of the saponins contain three or more glycosides. The following is the structural analysis of ginsenosides Rd and Rg1 as examples, and their fragmental pathway is shown in Figure 3.

The molecular formula of Rg1 was deduced as C_42_H_72_O_14_ with [M−H]^−^ at 799.4884. It can be inferred that the aglycone is connected to two molecules of glucose through glycosidic bonds according to the fragmental ions of 637.4348 [M−H−Glc]^−^, 475.3798 [M−H−2Glc]^−^, and 391.2859 [M−H−2Glc−C_4_H_12_]^−^. The structure of Rg1 was determined to be PPT-type ginsenoside according to the fragmental ions at 475.3798 [M−H−2Glc]^−^ and 391.2859 [M−H−2Glc−C_4_H_12_]^−^.

The molecular formula of Rd was deduced as C48H82O18 with [M−H]^−^ at 945.5496. It can be inferred that the aglycone is connected to three molecules of glucose through glycosidic bonds according to the fragments of 783.4966 [M−H−Glc]^−^, 621.4398 [M−H−2Glc]^−^, 459.3846 [M−H−3Glc]^−^. The type of Rd was determined to be PPD-type ginsenoside according to the fragmental ions at 375.2973 [M−H−3Glc]^−^, 391.2859 [M−H−2Glc−C_4_H_12_]^−^.

### 2.2. Multivariate Statistical Analysis of Samples

All of the 63 standard references were detected in four kinds of *P. ginseng* extracts. The TIC (total ion chromatogram) chart of the standard references and four kinds of samples was shown in Figure 4A,B. After a comprehensive analysis of the retention times, molecular weights, and ion fragments, a total of 261 ginsenosides including 54 potential new compounds from L5, 292 ginsenosides including 65 potential new compounds from L15, 273 ginsenosides including 59 potential new compounds from Y5, and 266 ginsenosides including 56 potential new compounds from N5 were identified, respectively.

In order to systematically evaluate the discrepancy of ginsenosides and identify their characteristic components in roots of *P. ginsengs* growing in four different environments, a workflow was constructed on the application of multivariate statistical techniques for further quantitative analysis.

Metabolomic-based strategy can be analyzed for semi-quantitative results at the same retention time due to its advantage of eliminating instrumental errors and correcting the retention time of the same compound between each sample. We investigated two analytical methods, PCA analysis and OPLS-DA [45]. A number of compounds that could influence the classification results were detected. The PCA and OPLS-DA analysis including four groups L5 (3 samples), L15 (2 samples), N5 (3 samples), and Y5 (3 samples), which was carried out with software SIMCA−14.0 (metrics) based on mass spectrometry semi-quantitative data. The total mass spectrometry data of several groups of samples were used for PCA and OPLS-DA analysis. Due to the existence of instrument errors, intra-group differences are inevitable. The type in the parameter of Scale was set to UV, and the Mode Type of analysis was set to OPLS-DA (supervised cluster analysis). In order to ensure the reliability of data analysis, all blank values will not be discarded. As shown in Appendix A, L5 and L15 performed similarly except for one sample in L5. Due to the irreducible differences between the samples, this PCA analysis convincingly reflects the differences between the samples in each group. With the exception of one sample that showed a clear separation, the sample parallelism of Y5 was recognized and shows a clear separation from L5, L15, and N5. The results for N5 were identical to those for Y5. Four groups of samples showed clear separation under supervision. As shown in Appendix A, the difference between the L5 and L15 groups, which is insignificant for N5 and Y5, is sufficient to distinguish L5 from L15. For the same species of ginseng, age has an effect on ginseng. L5 and L15 can be considered as two parts or as a whole distinct from N5 and Y5. This implies the influence of environment and age on ginseng. There was a small statistical difference between L5 and L15 due to different growth ages. It could be that L15 has more complex chemistry. However, it is worth noting that the difference between L5 and L15 was smaller than that between the other groups, which reminds us that growing environment has a greater influence on the chemical composition of ginseng than growing age. This can also be reflected in N5 and Y5. Although the two groups of samples are of the same age, in OPLS-DA analysis, they show great differences due to different growing environments. As shown in Appendix A, intra-group sample parallelism was trustworthy, with four groups of samples showing clear separation under supervision. Unsupervised PCA analysis retained the original data of the samples and clearly showed the differences between the samples. In the premise that PCA analysis is accepted, supervised OPLS-DA analysis will show more intuitive separation. In the cluster analysis, there were obvious differences among the four groups, which fully reflected the influence of growing environment and growing age on the chemical composition of ginseng.

To further verify the accuracy of the result of PCA and OPLS-DA, semi-quantitative results of 312 identified ginsenosides was further submitted for heat map analysis (Figure 5A). The result showed that L5 and L15 could be grouped together due to the similarity of ingredients, while there was still a difference between these two groups on the color depth, which could be speculated that the growth time played an important role in the quantity of *P. ginseng* [3]. Y5 and N5 can be divided into another group, indicating that N5 and Y5 have a greater similarity compared with the forest-growing ginseng. The results showed that the growth environment plays an important role in the growth process of ginseng.

The Venn diagram of 312 *P. ginseng* saponins in the four group samples was shown in Figure 5B. Among the detected ginsenosides, 221 kinds of *P. ginseng* saponins were shared by the four groups of *P. ginseng*, while 9 ginsenosides were unique in L15 which was significantly higher than others. There were 6 unique ginsenosides in N5, while only 1 unique ginsenoside were detected in L5 and Y5. The results suggested that the variety of chemical composition in *P. ginseng* is closely related to its growth period.

### 2.3. Discussion

Semi-quantitative contents of 312 ginsenosides are shown in the Appendix A. The S-Plot analysis for the pairwise comparison of the four groups of samples is shown in Appendix A, and there are obvious analyte differences among the groups, which validates the conclusion that the influence of growth environment on chemical constituents in ginseng. In S-plot analysis, the Type in the parameter of Scale was set to Par, and the Mode Type of analysis was set to OPLS-DA (supervised cluster analysis) to ensure the reliability of data analysis. As far as ginsenosides are concerned, the differences from S-plot analysis between the groups were attributed to the unique ginsenosides in the four groups of samples, such as Yesanchinoside E/isomer (L5, rt = 5.19), new ginsenosides 3 (N5, rt = 1.48), new ginsenosides 12 (N5, rt = 2.36), 20(R)-G-Rf2 isomer (N5, rt = 5.28), 20(R)-G-Rf2 isomer (N5, rt = 8.69), G-Rh14 isomer (N5, rt = 10.78), G-Rh4 isomer (N5, rt = 19.07), Diacetyl-G-Rd isomer (Y5, rt = 6.26), new ginsenosides 5 (L15, rt = 1.54), Vinaginsenoside R6 isomer (L15, rt = 3.30), 3β,12β,20,25-tetrahydroxydammarane-6-*O*-β-d-xylopyranosyl-(1→2)-β-d-glucopyranoside isomer (L15, rt = 4.15), new ginsenosides 25 (L15, rt = 5.62), chikusetsusaponin LM2 isomer (L15, rt = 6.64), new ginsenosides 38 (L15, rt = 7.45), new ginsenosides 40 (L15, 7.84), ZB-RI-6‘ methyl ester (L15, rt = 18.44), Rs4 (L15, rt = 18.44). In addition, ginsenosides with large differences in content also contributed greatly to the difference between groups, such as Pseudoginsenoside Rs1/isomer, G-Re2, G-Re1 isomer, NG-R1, G-Re. In L15 and Y5, the highest content of ginsenoside was G-Rg1, the contents of G-Rg1, G-Ro, G-Rb1, and 20-G-Rf are the top four compounds. While in N5 and L5, the highest content of ginsenoside was G-Re, the top four compounds are G-Re, G-Ro, G−Rb1, and 20-G-Rf. The L15 *P. ginseng* has the most diversity of ingredients in the four groups in terms of the total number and unique types of ginsenosides. To a certain extent, its chemical composition was most similar to L5 *P. ginsengs* and had significant differences from the other two groups of *P. ginsengs*, according to the results of PCA and OPLS-DA analysis. Comparing the difference of chemical components between L5 and L15 *P. ginseng*, the identified differential compounds were found lower in L5 *P. ginseng* by semi-quantitative analysis, which is presumably due to the transformation of the related components in L5 to other compounds in L15 after a long growth time. Compared with Y5, 42 different ginsenosides were detected in L15 *P. ginseng*. Except for the differential growth time, it was speculated that different growth environments took an important role in the formation of compounds especially the change of trace components in *P. ginseng*, and 58 different ginsenosides were found between N5 and L15 *P. ginseng* [46].

The N5 *P. ginseng* grew in farmland, while Y5 P. ginseng grew in a more natural area in mountains. Although having the same growth period, there were 44 different components between *P. ginsengs* of N5 and Y5. Thus, the growth environment, as moisture, air, and trace elements, might be the main reason for the difference between them. In terms of the growth years, L15 could better reflect the diversity and specificity of ginsenosides than L5. This shows that the growth years and ginsenoside types show a positive correlation. Y5 and N5 have similar exposure times and humidity, the only difference between them is that Y5 is at a higher altitude which results in lower temperature for Y5 than for N5. It can be concluded that the low temperature environment is conducive to the development of ginsenoside diversity. L5 is surrounded by forests and receives less sunlight, resulting in a long-term humid environment for L5; however, these are absent in N5. It can be seen that the moist, cool and low temperature environment provides a positive support for the amount of saponins in ginseng [47].

The proportions of different types of ginsenosides in the four groups of *P. ginseng* are shown in Figure 5C. The PPD-type and PPT-type ginsenosides account for the largest proportion of ginsenosides in our research, similarly to relevant studies on *P. ginseng* [48]. It showed that protopanaxadiol and protopanaxatriol saponins were the main constituents in *P. ginseng*, and the proportions of different ginsenoside types are similar in the four groups. Except for the protopanaxadiol saponins, not only the quantity of total saponins but also each other types saponins of *P. ginsengs* in L15 were higher than that in other groups. Comparing L15 with L5, it was concluded that the growth time was an important cause of the chemical difference of *P. ginseng* under the same environmental conditions. As for the difference between L5, Y5, and N5, it can also be concluded that the ecological environments also play a key role in the growth process of ginseng.

## 3. Experimental

### 3.1. Materials

Three batches of five-year-old *P. ginseng* cultivated in the forest (ginseng is grown in forest environments similar to wild ginseng.) (L5), two batches of fifteen-year-old *P. ginseng* cultivated in the forest same as L5 (L15), three batches of five-year-old *P. ginseng* cultivated in mountains (ginseng is grown on hill farms that have been cleared of previous plants and made manageable) (Y5), and three batches of five-year-old *P. ginseng* cultivated in farmland (ordinary farmland in the plains) (N5) (Appendix A) were collected for analysis. The differences among the three growth environments were natural factors such as temperature, humidity, and sunlight. Intervention factors such as fertilization and watering remained the same. All samples were from Ji’an City, Jilin Province, China, and identified by Professor Xiu-Wei Yang of the School of Pharmaceutical Sciences of Peking University. The voucher specimens were deposited in the State Key Laboratory of Natural and Biomimetic Drugs of Peking University.

### 3.2. Standard Samples, Chemicals, and Reagents

Sixty-three kinds of ginsenosides of *P. ginseng* were used for qualitative analysis: G-Re3 (1), 20-glu-G-Rf (2), G-Re4 (3), G-Re2 (4), NG-R1 (5), G-Re1 (6), G-Rg1 (7), G-Re (8), G-Ro (9), NG-R4 (12), 20(S)-G-Rf-1a (13), 20(S)-G-Rf (14), 20(R)-G-Rf (15), 20(S)-NG-R2 (16), G-Ra2 (18), G-Ra3 (19), 20(R)-NG-R2 (20), G-Rb1 (21), 20(S)-G-Rg2 (22), 20(R)-G-Rg2 (23), G-Ra1 (24), G-Rc (25), 20(S)-G-Rh1 (26), 20(R)-G-Rh1 (27), G-Rb2 (28), G-Rb3 (29), Q-R1 (30), G-Rd (31), G-Rk1 (32), Rs2 (33), Rs1 (34), G-Ro methyl ester (35),G-Rh19 (36), DHDGG (37), G-Rg6 (38), G-F4 (39), DHDXG (40), CS-Iva methyl ester (41), G-Rk3 (42), G-Rh4 (43), G-Rg9 (44), G-F2 (45), 20(S)-G-Rg3 (46), 20(R)-G-Rg3 (47), DEDT (48), 20(S)-PPT (49), 20(R)-PPT (50), 20(S)-G-Rs3 (51), 20(R)-G-Rs3 (52), Rs4 (53), ZB-RI-6′ methyl ester (54), G-Rg5 (55), 20(S)-G-Rh2 (57), 20(R)-G-Rh2 (58), DDT (59), 20(S)-PPD (62), and 20(R)-PPD (63). Among them, CS-IV (10), CS-Iva (11), G-F3 (17), G-C-K (56), G-Rh3 (60), and G-Rk2 (61) were purchased from Chengdu Pusi Biotechnology Co., Ltd., and the other ginsenosides were from the reference’s library of our research group. The purity of each reference compound was more than 98.0% determined by RP-HPLC. The chemical structures of the above 63 compounds were shown in Appendix A.

LC-MS grade acetonitrile (ACN), methanol (MeOH), and ammonium formate were purchased from Thermo Fisher Scientific (Fair Lawn, NJ, USA). The deionized water was prepared by a Millipore Alpha-Q water purification system (Bedford, MA, USA).

### 3.3. Subsubsection Instruments Chromatographic Conditions and Parameters

SCIEX Triple TOF 6600+ was accompanied with the SCIEX Exion LC AD system. The mass spectrometry parameters were as below: auxiliary gas (GS1): 60 psi; atomizing gas (GS2): 60 psi; curtain gas (CUR): 35 psi; ion source temperature (TEM): 600 °C; ion spray voltage (IS): positive ion 5500 V, negative ion −4500 V; declustering voltage (DP): 80/−80 V; collision energy (CE): 40 ± 20 V. The software SCIEX OS combined with Peakview was used to process the data. Composition of the eluent was selected by multi-dimensional investigation of sample resolution, peak shape, and sensitivity. After investigating different elution proportions such as H*2*O-acetonitrile, 0.1% formic acid–acetonitrile, 0.5 mmol/L ammonium formate–acetonitrile, the eluent made up with 0.5 mmol/L ammonium formate (A)–acetonitrile (B) was finally selected to analyze the samples. A programmed ingredient was carried out as follows: 0–3 min 78% A; 3–5 min 78–70% A; 5–9 min 70–65% A; 9–11 min 65–60% A; 11–13 min 60–52% A; 13–18 min 52–45% A; 18–22 min 45–30% A; 22–25 min 30–10% A; 25–26 min 10% A. An ACQUITY UPLC^®^ CSH (C18 1.7 μm, 2.1 × 00 mm; Waters) coupled with an AC QUITY UPLC^®^ BEH Shield RP 18 1.7 μm pre-column was used for chromatographic separation. The injection volume was 2 μL. The flow rate was 0.35 mL/min. The temperature of the column oven was kept at 40 °C, while the temperature of the autosampler was kept at 25 °C.

MS scan conditions were optimized both in negative and positive ion modes by using reference standards. Due to more peaks and much clearer fragmental information, the negative ion mode was chosen to analyze most of the ginsenosides, while compounds 49 and 60 could be observed only in positive ion mode. For negative ion mode, the [M−H]^−^ or [M+FA−H]^−^ was shown as molecular ion peaks, and the molecular ion peaks for 49 and 60 were [M+H]^+^ in positive ion mode. The base peak chromatographic profiles of the standards and samples are shown in Figure 1.

### 3.4. Preparation of Samples and References Standards

After being washed with water, the roots of *P. ginseng* were dried at 40 °C for 40 h. Then the materials were crushed to powder and filtered with a 60-mesh sieve. The 0.5 g fine powder of each sample (60 mesh size) was ultrasonic extracted (250 W, 40 kHz) with 20 mL of 70% MeOH (*v*/*v*) for 30 min. The samples were then set aside to room temperature and refilled the lost weight with 70% MeOH. After centrifugation at 3000 rpm for 10 min, the supernatant was passed through a 0.2 microns filter membrane to obtain the tested solution and stored at −20 °C for analysis. Each reference was precisely weighed and dissolved with MeOH to form the final concentration at 1 mg/mL. An appropriate volume of each reference solution was mixed to obtain the working solution for qualitative analysis.

### 3.5. Statistical Analysis

Orthogonal partial least-squares discriminant analysis (OPLS-DA) was used for analysis with SIMCA-P software 14.0 (Umetrics, Umea, Sweden). The semi-quantitative content of 320 ginsenosides in the four groups samples were compared by hierarchical clustering analysis (the website at https://www.metaboanalyst.ca/, accessed on 5 September 2022). The number of saponins in the four groups was used to produce Venn diagram. GraphPad Prism 7 was used to visually display the proportion of each type of ginsenosides in the four groups.

## 4. Conclusions

This study optimized the method of extracting complicated compounds from *P. ginseng*, and established a sensitive chromatographic method for components separation. Using positive and negative ion modes, 320 kinds of ginsenosides were identified in *P. ginseng* by Triple-TOF-MS, in which 75 ginsenosides were identified as potential new compounds. The chemical structures of ginsenosides were initially verified by systematic elucidation of the accurate mass, retention times, and fragment ions. The cluster analysis showed that the growth environment had a great influence on the chemical composition of *P. ginseng*. The discovery of this study allowed us to further understand the pharmacodynamic substances and growth habits of *P. ginseng*, and will provide a reference to study on quality evaluation of *P. ginseng* in the future.

## Figures and Tables

**Figure 1 molecules-28-02068-f001:**
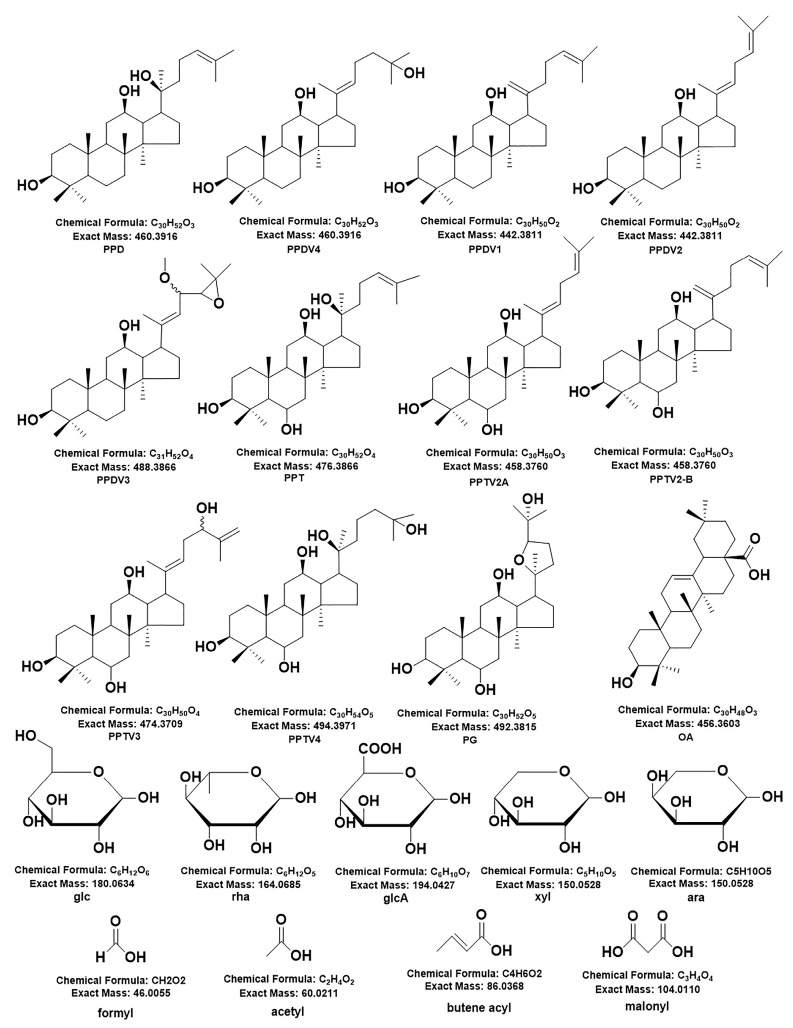
The 75 potential new ginsenosides’ type of aglycone and glycone. PPDCV and PPTCV represent PPD type and PPT type saponins with side chain changes, respectively.

**Figure 2 molecules-28-02068-f002:**
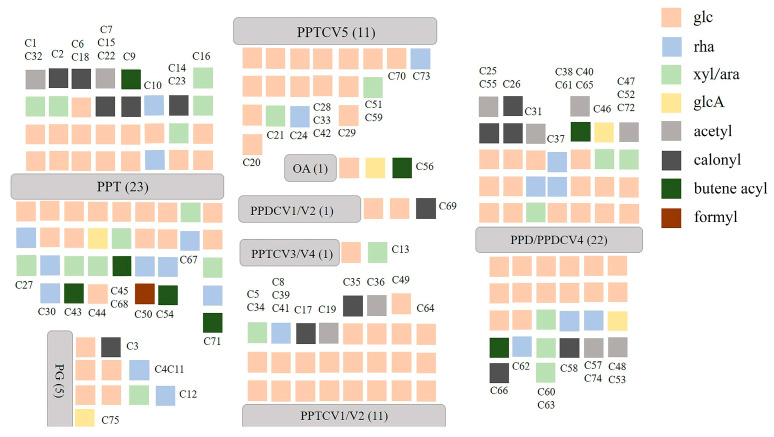
Structure analysis of 75 potential new ginsenosides.

**Figure 3 molecules-28-02068-f003:**
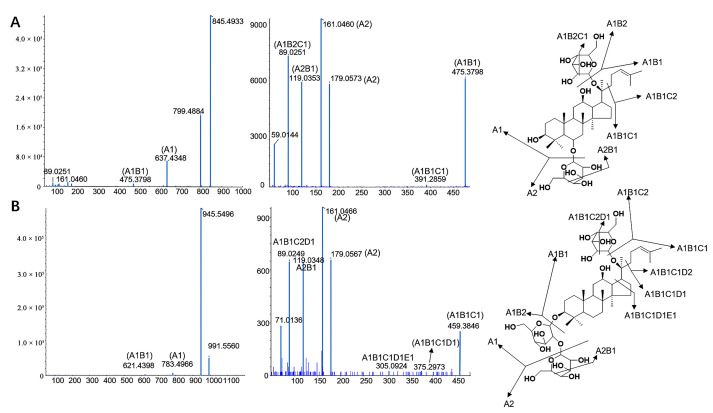
The MS/MS spectra and fragmentation process of Rd (**A**) and Rg1 (**B**).

**Figure 4 molecules-28-02068-f004:**
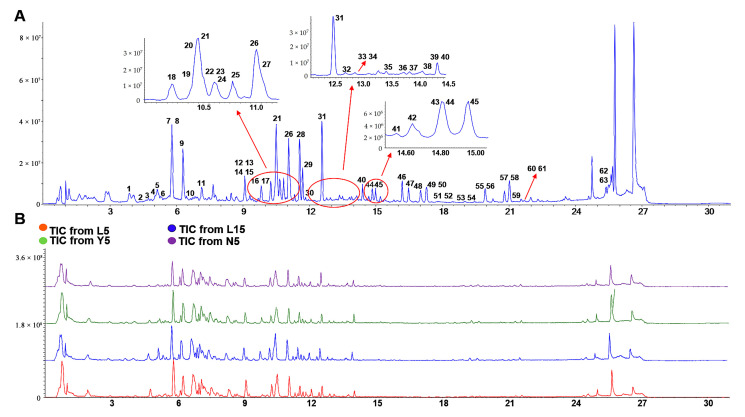
TIC of the ginseng standard references and four ginseng samples. (**A**) TIC of the ginseng standard references. (**B**) TIC of the four ginseng samples.

**Figure 5 molecules-28-02068-f005:**
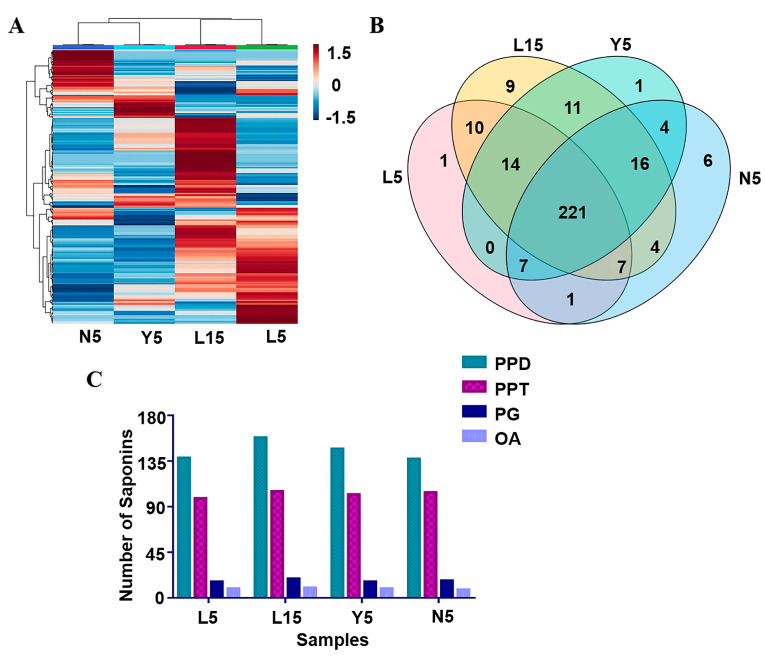
Multi-statistical analysis of the components in four growth environments ginseng samples. (**A**) Heat map of four growth environment ginseng samples. (**B**) The Venn diagram of the 312 *P. ginseng* saponins. (**C**) Proportion of different type ginsenosides in the four growth environment ginseng samples.

## Data Availability

Not applicable.

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
