# Peer review of "Characterization of Ginsenosides from the Root of Panax ginseng by Integrating Untargeted Metabolites Using UPLC-Triple TOF-MS"

_molecules, 2023, doi:10.3390/molecules28052068_

Round 1

Reviewer 1 Report (Previous Reviewer 2)

All the points raised were addressed and the manuscript is improved. However, there are still many typos. The authors have to pay more attention to the details. Although they are minor and are not scientific parts, most reviewers may question the scientific potentials.

Negative signs for ions and some of the formulas have to be corrected. Please see attached file where highlights indicate them.

The added information in lines 141–144 are okay but the English is not. The also the file.

Author Response

Comment: All the points raised were addressed and the manuscript is improved. However, there are still many typos. The authors have to pay more attention to the details. Although they are minor and are not scientific parts, most reviewers may question the scientific potentials.

Response: Thank you for your comments. We have checked the article and revised some mistakes.

Comment: Negative signs for ions and some of the formulas have to be corrected. Please see attached file where highlights indicate them.

Response: Thank you for your question, and we have revised these formatting errors.

Comment: The added information in lines 141–144 are okay but the English is not. The also the file.

Response: The language at this section has been revised.

Reviewer 2 Report (New Reviewer)

 First, I was very surprised to see this paper, there is not a single reference in the discussion part of the whole paper. However, there are 45 references in the references section, but why aren't there so many cited references in the text.

Second, the logic of the whole paper is unclear, and there is no detailed description of the influence of different cultivation environment and growth years on the ginsenosides from the root of Panax ginseng in Introduction and discussion. This paper should focus on the influence of different cultivation conditions and growth years on the ginsenosides from the root of Panax ginseng, rather than simply stating the difference of the ginsenosides from the root of four ginseng.

Third, the P. ginseng In the whole paper should be italic.

Fourth, the authors should elaborate on the cultivation conditions at the three planting sites (1: nature forest in mountain; 2: mountains after logging, removing the 74 root system; and 3: farmland ordinary farmland in the plains), such as fertilization, as different fertilization conditions can also seriously affect the ginsenosides from the root of Panax ginseng.

Author Response

Comment: First, I was very surprised to see this paper, there is not a single reference in the discussion part of the whole paper. However, there are 45 references in the references section, but why aren't there so many cited references in the text.

Response: Thank you for your question, many references are used to support the identified compounds. In addition, according to your comments, We have added some references in the discussion section.

Comment: Second, the logic of the whole paper is unclear, and there is no detailed description of the influence of different cultivation environment and growth years on the ginsenosides from the root of Panax ginseng in Introduction and discussion. This paper should focus on the influence of different cultivation conditions and growth years on the ginsenosides from the root of Panax ginseng, rather than simply stating the difference of the ginsenosides from the root of four ginseng.

Response: According to your comments, we have introduced the important relationship between ginseng and environment in the introduction section and expanded the discussion section.

In terms of the growth years, L15 could better reflect the diversity and specificity of ginsenosides than L5. This shows that the growth years and ginsenoside types show a positive correlation. Y5 and N5 have similar exposure times and humidity, the only difference between them is that Y5 is at a higher altitude, which results in lower temperature for Y5 than for N5. It can be concluded that the low temperature environment is conducive to the development of ginsenoside diversity. L5 is surrounded by forests and receives less sunlight, resulting in a long-term humid environment for L5, however, these are absent in N5. It can be seen that the moist, cool and low temperature environment provides a positive support for the amount of saponins in ginseng.

Comment: Third, the P. ginseng in the whole paper should be italic.

Response: All of P. ginseng in the manuscript have been revised to italic.

Comment: Fourth, the authors should elaborate on the cultivation conditions at the three planting sites (1: nature forest in mountain; 2: mountains after logging, removing the 74-root system; and 3: farmland ordinary farmland in the plains), such as fertilization, as different fertilization conditions can also seriously affect the ginsenosides from the root of Panax ginseng.

Response: According to your comments, we have described the living environment of the sample in detail in the material section.

Round 2

Reviewer 2 Report (New Reviewer)

The author has modified the content according to the requirements, which can be accepted.

This manuscript is a resubmission of an earlier submission. The following is a list of the peer review reports and author responses from that submission.

Round 1

Reviewer 1 Report

You still do not show neither PCA results nor quality parameters that would help to judge the soundness of your analysis. I already pointed you to the article byWorley and Powers (https://www.ncbi.nlm.nih.gov/pmc/articles/PMC4990351/) that clearly reveals that OPLS-DA ALWAYS gives good separation regardless whether this is true or not.

The article has not improved at all.

Reviewer 2 Report

The manuscript “Metabolomic-based strategy for chemical constituents’ identification of Panax ginseng Meyer in different growth environments” submitted by Sun et al. describes the analysis and verification method of Panax ginseng Meyer by statistical data treatment of LC-MS spectra of extracted ginsenosides. The manuscript contains important and useful information and is well organized. Some comments are listed below to improve it.

Throughout the manuscript:

1) There are many words those separated and hyphenated. They must be fixed.
Lines 106–108:inves-tigation, elu-tion, ammo-nium; Line 118: refer-ence, lines 158, 159...so many to mention.

2) Make numbers in H2O and formula like C31H51O4 subscript.

3) Italicize “m/z”.

4) Change "glc, rha, xyl etc." to "Glc, Rha, Xyl etc".

Line 19: …to characterize the ginsenosides extracted by ultrasonic from P. ginseng… —>…to characterize the ginsenosides obtained by ultrasonic extraction from P. ginseng…  

Line 52–53: The authors noted that “many analytical methods” have problems in sensitivities, however, the techniques followed are different categories of those. For example, methylation alone is not used for characterization thus coupled with other analytical methods such as HPLC and MS etc. The sentence has to be revised.

Line 76: Is this a designated protocol for certifying the type of product in China? A review do not understand thus readers around world do not as well. If it is accepted way of way of writing in the Molecules or any other journals, it is okay. A reviewer has never seen it before.

Line 95: Standard compounds 1–11 were missing in Table S2.

Line 128: degree is underlined.

Line 141–143: Explanation required for metaboanalyst.ca and omicshare.com .

Line 158 and 201: Change “The type of aglycone and glycosyl are shown in…” to “The type of aglycone and glycone are shown in ”.

Line 210–214: I understand some common compounds exists among potentially new compounds in individual samples. It does not have to be so stressed on the new compounds for me, but if the authors want to stress, I suggest to include Venn diagram as shown in Fig. 5 as SI.

Line 224–225: …analysis of ginseng chemical composition in different latitude and longitude, … This may be true, but it sounds over stressed, because samples were collected within a province. If the authors want to stress it, more samples from different areas have to be investigated.

Line 234: L15 and L5 could be roughly sorted into one group because…  A reviewer disagree what the authors say. The location of sampling may be close each other, but the data in Fig. S2 and others indicate they can be distinguished. This statement is inconsistent with Fig. 5 and sentence in line 303 as well. Revision needed.

Line 306: the ecological environments also play a key role in…  It might be true, however, growth environment just affects their metabolism...Is this what the authors wanted to say? It may relate to quality control reasons of Chinese medicine or more scientifically the cause of the changes. Investigation of the latter may be of extreme difficult but important issue. On the contrary, the former is easy to comment and is acceptable I suppose.

Line 309: The legend of Fig. 5 is screwed. Revise.